# Boiling Heat Transfer Characteristics of Porous Microchannel with Pore-Forming Agent

**Qinhui Lei, Donghui Zhang \*, Lei Feng, Jijin Mao and Daifen Chen**

School of Energy and Power Engineering, Jiangsu University of Science and Technology, Zhenjiang 212003, China
\* Correspondence: dhzhang20@126.com; Tel.: +86-15905287879

**Abstract:** Traditional microchannel needs to face the flow-reversal difficulty in high heat fluxes due to limited space. It results in large pressure and temperature fluctuation. Porous microchannels arouse more interest to provide a new solution to this problem. Flow boiling experiments in porous microchannels with PFA were investigated. Porous microchannels were sintered by 10 μm (or 30 μm) spherical copper particles with pore-forming agent ($Na_2CO_3$, 60–90 μm). Porous microchannels were composed of 23 parallel porous microchannels with 600 μm in width and 1200 μm in depth. The addition of PFA (pore-forming agent) could increase the sample porosity. For Q10 series, sample porosities increase from 20.4% to 52.9% with the PFA percentage change from 0% to 40%, while for the Q30 series they increase from 26.6% to 47.5%. Experimental results showed the boiling heat transfer coefficient (HTC) reached the maximum at the moderate porosity for both Q10 and Q30 series. Too large or too small porosity would degrade boiling heat transfer performance. It demonstrated that there existed an optimal range of PFA content for sintered microchannels. PFA content has a minor effect on the average pressure drop and would not cause the rapid increase in flow resistance. Visual observation disclosed that the sample porosity would affect the pressure instability significantly. The sample with moderate porosity showed periodic pressure fluctuation and could establish rhythmical boiling. Particle size also exerted a certain influence on the boiling heat transfer performance. Q30 series could achieve higher HTC and CHF (Critical heat flux) than Q10 series. This is attributed to the larger ratio of layer-thickness-to-particle-size ($\delta/d$) for Q10-series samples.

**Keywords:** flow boiling; microchannel; pressure fluctuation; pore-forming agent; sintering

## 1. Introduction

The problem of heat dissipation has become the main obstacle of product upgrading. The heat flux in local hot spots exceeds even more than 200 W/cm². Multifaceted requirements are considered in the design of a heat dissipation system: high efficiency, miniaturization, low cost, energy saving, etc. Traditional air and single-phase liquid cooling methods are transformed to the phase-change one. Since the 1980s, the microchannel boiling system has attracted extensive attention, as it dissipates the heat by the two-phase flow boiling process. It combines various advantages of high HTC and low working-fluid demand [1,2]. Due to limited space, traditional microchannels need to face flow reversal difficulty in high heat fluxes, which results in large pressure and temperature pulsation [3,4]. In recent years porous microchannels have attracted more interest to provide a new solution to this problem [5,6].

The porous microchannel could be sintered by different approaches, such as the direct-sintering method [7], electrochemical deposition method [8], pore-forming agent method [9] and so on. Among them, the advantage of the pore-forming agent method lies in its effective manipulation on porosity and pore size distribution. It may be helpful to alleviate the contradiction between capillary force, liquid supply, and bubble detachment in high heat fluxes. Xu and Zhou [9,10] made bi-porous nickel wicks in Looped heat pipes (LHP). They combined nickel powder with the pore-forming agent (Pure NaCl). Sensitive

analysis of five sintering parameters was conducted on porosity, permeability, capillary pumping head and effective thermal conductivity. They found the PFA content is the largest effect factor and the particle size of PFA was relatively small. Sintered samples are highly sensitive to the PFA content. Yan and Xu [11,12] used the anhydrous $Na_2CO_3$ as a PFA to produce a kind of micro-nano capillary wick. They found that multiscale structure could reduce the evaporator wall temperature greatly compared with the microchannel/wick evaporator. Small pores (µm scale) provide great capillary force for liquid suction, large pores (10 µm scale) between clusters increase surface area for liquid film evaporation, and microchannels (mm scale) are responsible for vapor venting. Lin [13] applied the two-steps sintering method to build an excellent bi-porous structure: the first step is to form perforated clusters by mixing Nickel powder with the binder, and the second to sinter these packed clusters to complete the bi-disperse wick. Its maximum heat transfer coefficient could reach 23.3 $kW/m^2·K$, which is approximately 230% of the mono-porous wick at heat load of 400 W. Liu and Kandlikar [14] combined the drop coating and screen-printing technique with the sintering method. $Na_2CO_3$ was added to alter the porosity of the porous layer. Some samples could achieve 303 $W/cm^2$ in CHF in the pool boiling test. The effect of coating thickness on pool boiling performance was also investigated. However, this work did not present detailed porosity and PFA content. Wei and Yang [15] used both the loose sintering (LS) method and cold pressure sintering (CPS) method to investigate the PFA content on LHP evaporator performance. They found that the porosity and average pore size increased with the PFA content, and capillary performance was enhanced due to the improvement in the connectivity of internal pores. The optimal PFA content was suggested to be <10% for loose sintering and <20% for cold press sintering.

From the current research, the pore-forming agent method was mostly applied in capillary-wick fabrication of heat pipes, but less investigated on flow boiling direction. The flow boiling process in the sintered microchannel was confronted with the 'porosity confliction' problem: if the porosity is too low, the nucleation site density and heat transfer area will decrease, and the vapor discharge resistance would increase; and if too large, the effective thermal conductivity resistance increases though the vapor departure resistance decreases. It indicated there existed an optimum porosity for porous microchannels. For the porous microchannel, the sample performance is influenced by various structural parameters: particle size, porosity, and pore morphology, etc. These parameters are coupled together. The PFA method is helpful to illuminate the porosity effect independently in the porous microchannel boiling process.

This study focused on the PFA content influence on the flow boiling performance of porous microchannels. Two kinds of spherical copper particle size, 10 µm and 30 µm, were investigated. Subcooled flow boiling experiment was conducted with the deionized and de-gassed water as working fluid. The influence of sample porosity on the boiling heat transfer and pressure drop was emphatically investigated. To understand the boiling process in depth, visual observation is applied in combination with pressure synchronous acquisition.

## 2. Experimental System

### 2.1. Experimental System

The experimental system is mainly composed of pipe unit, electric heating unit and data-acquisition unit, as shown in Figures 1 and 2. The pipe unit includes the deionized-water tank, micro gear pump, flowmeter, microchannel testing section, and terminal collection reservoir. After cooling to a preset temperature, the degassed and deionized water in the tank is driven to the microchannel test section by a micro gear pump (MG200XK/DC24W). In the sintered porous microchannel the liquid water is converted into the liquid–vapor two-phase state, and finally collected by the terminal collection reservoir. A copper block is used as the simulated heat source. Six electric-heating rods (6 × 250 W) are powered by an adjustable DC power supply (Voltage range: 0~300 V; Current range: 0~10 A). The flow rate was measured by a micro-turbine-type flow meter (FHK 937-1510), ranged from 1 to 10 L/h, with an uncertainty of ±1%. Liquid temperatures of inlet and outlet of test section are measured by

the two PT100 sensors (I-class accuracy, $\pm 0.3$ K), respectively. Temperature measurements of the copper block are conducted by armored T-type thermocouples (I-class accuracy, $\pm 0.4$ K). Two pressure sensors, PX309-015G5V, are responsible for the measurement of inlet and outlet pressure, with a range of 0~15 psia (Relative pressure). The pressure data are gathered by the NI PCI-6212 module, and temperature data by Agilent 34901A. All measurement data are transferred to the computer for real-time display. Phantom VEO E-310L is applied to capture the boiling flow pattern at an image capture rate of 2800 frames/s and resolution of 1280 × 800 pixels. Synchronous acquisition of pressure sensor and high-speed camera is performed to acquire the dynamic evolution of flow pattern in detail. Before each test, Nitrogen gas should be filled into the test section to check its sealing degree.

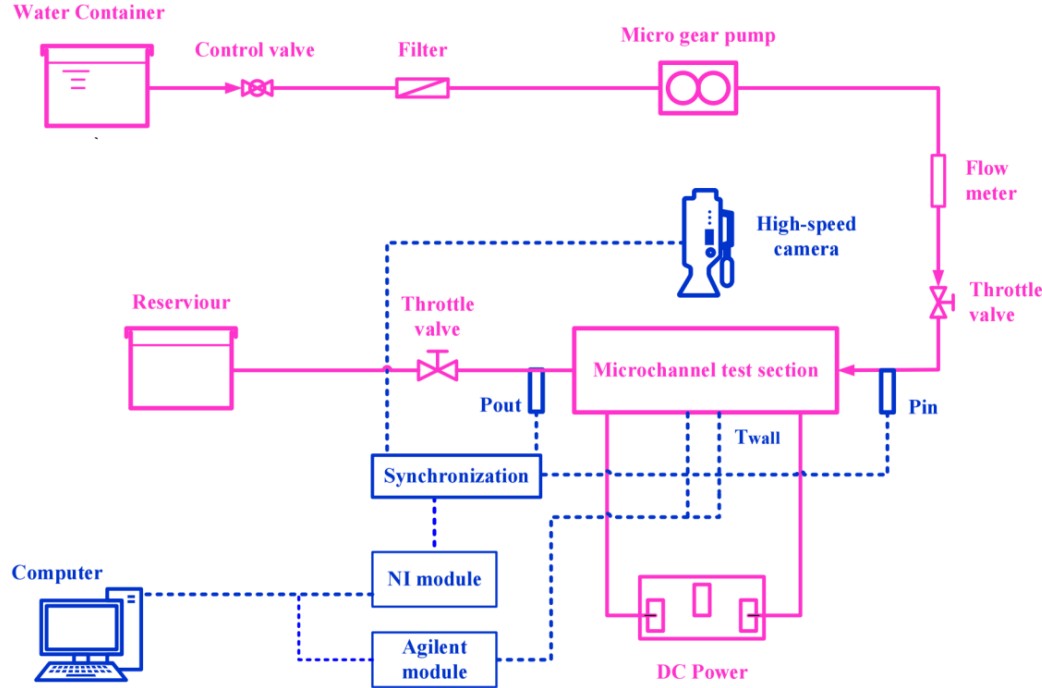

**Figure 1.** Flow diagram and measurement system.

The microchannel test section in Figure 3 mainly includes a transparent PC cover plate, PEEK chamber, copper block and glass-fiber adiabatic sheath. The upper part of the copper block is designed as a rectangular copper column to ensure one-dimensional thermal conductivity (three temperature-measuring holes are arranged longitudinally along the center line to monitor the input heat flux), and six heating-rod holes are evenly arranged in the lower part. The exterior of the copper block base is wrapped with glass-fiber sheath, which greatly reduces the heat loss of the test section. The schematic diagram is described in Figure 4 about the sintered porous microchannel and copper plate. Structural parameters of the porous microchannel are displayed in Table 1. The footprint area of each sintered sample is 28 × 28 mm. It is composed of 23 microchannels. Each microchannel has a channel width of 600 μm, fin width of 600 μm and channel depth of 1200 μm. The porous microchannel sample is firstly sintered onto the copper plate surface (thickness: 5 mm) in the sintering furnace under the protection of Nitrogen gas.

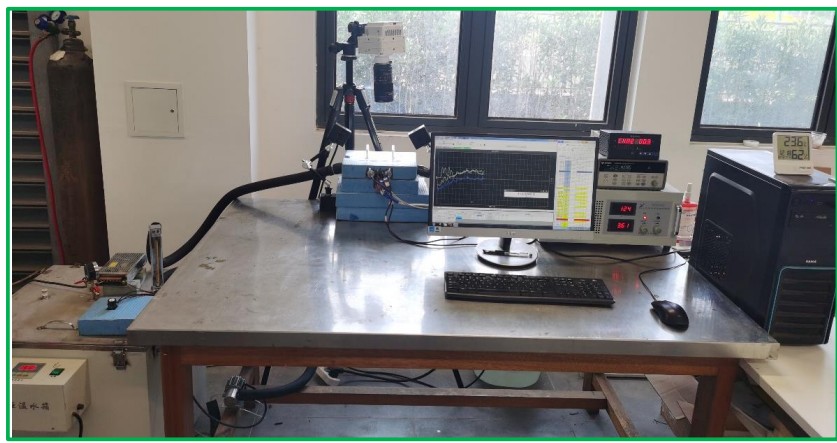

**Figure 2.** Schematic diagram of experimental setup.

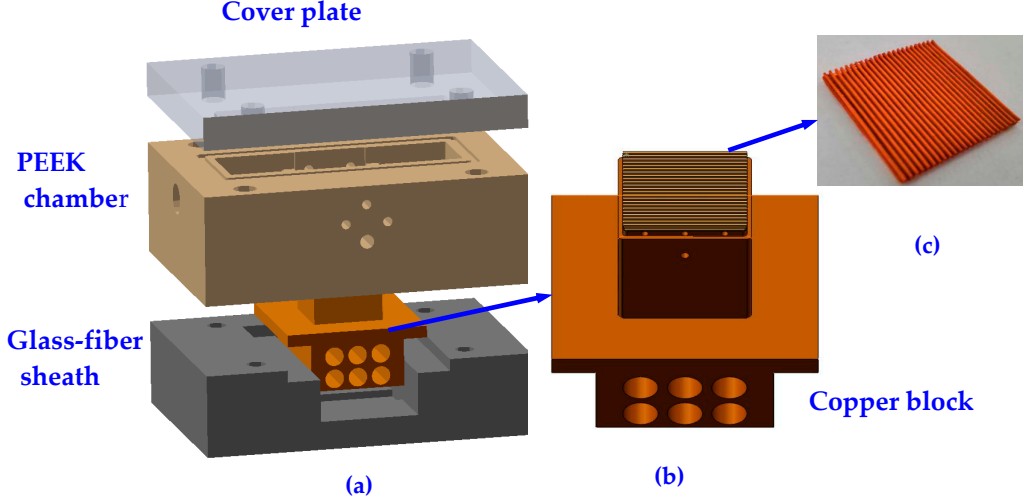

**Figure 3.** Microchannel test section: (**a**) Microchannel test section; (**b**) Copper block; (**c**) Porous microchannel.

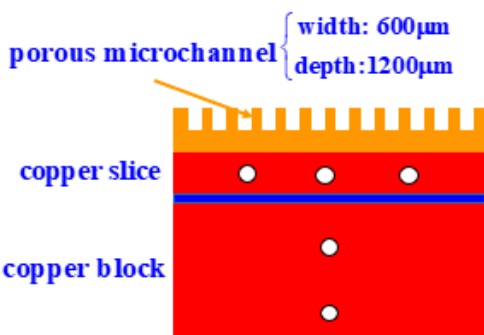

**Figure 4.** Schematic diagram of sintered porous microchannel and copper plate.

**Table 1.** Structural parameters of porous microchannels.

| Width (μm) | Depth (μm) | Layer Thickness (μm) | Inter-Rib Width (μm) | Channel Number |
|---|---|---|---|---|
| 600 | 1200 | 200 | 600 | 23 |

Then, the copper plate is soldered in the upper surface of copper block by the SAC305 soldering slice (Sn–Ag–Cu, λ = 48 W/m·K, thickness: 0.1 mm) in the vacuum state. Struc-

tural parameters of porous microchannels are listed in Table 1. The wall temperature of the copper plate is extrapolated from the thermocouples in the copper plate. Three measuring holes are lengthwise arranged along the copper plate and the other two vertically arranged in copper block, as shown in Figure 3. The gap between the sample and test chamber is filled by the high-temperature sealant.

### 2.2. Fabrication of Porous Microchannel and Characterization

The porous microchannel is sintered by mixing 10 μm (or 30 μm) spherical copper particle with pore-forming agent $Na_2CO_3$ (60–90 μm), as shown in Figure 3. Before the sintering process the copper powder and $Na_2CO_3$ were screened separately, and then mixed with a certain volume ratio. The mixed powder is evenly shaken and filled into the graphite mold. Finally, these samples are sintered under the protection of hydrogen and nitrogen atmosphere. The sintering temperature is 850 °C, which is also the melting temperature of $Na_2CO_3$. After cooling to room temperature the sintered sample is placed in a water bath for 2 h to dissolve the melted Sodium carbonate.

Here, 'particle size-PFA content' is used to distinguish each sample by shorthand. For example, Q30-20% represents the sample sintered from 30 μm copper particles with 20% PFA content. Sample porosities are illustrated in Tables 1 and 2, measured by Archimedes' method.

**Table 2.** Parameters of microchannels sintered with 10 μm spherical copper powder.

| Particle Size | 10 μm | | | |
|---|---|---|---|---|
| specification | Q10-0% | Q10-10% | Q10-20% | Q10-40% |
| volume content | 0% | 10% | 20% | 40% |
| porosity | 20.4% | 29.6% | 39.1% | 52.9% |

Figure 5 shows the micro-structures of both Q10-series samples under different PFA contents, obtained by GeminiSEM 360. Compared with Q10-0%, Q10-40% not only increases the porosity of the sample, but also forms a kind of dual-pore morphology: large pore more than 50 μm, and small pore below 10 μm. Some large PFA particles, after clearing out, create large voids inside the sample, as marked with dotted green circle. Some copper particles agglomerate together to form a cluster structure. Micro-structure images of both Q30-series samples are compared in Figure 6. A large void structure is also commonly observed, but not as obvious as Q10-40%. Cluster structure is difficult to be sought out. This is attributed to the proximity between the PFA particle size and copper particle size. The PFA sintering method increases the porosity and alters pore morphology certainly.

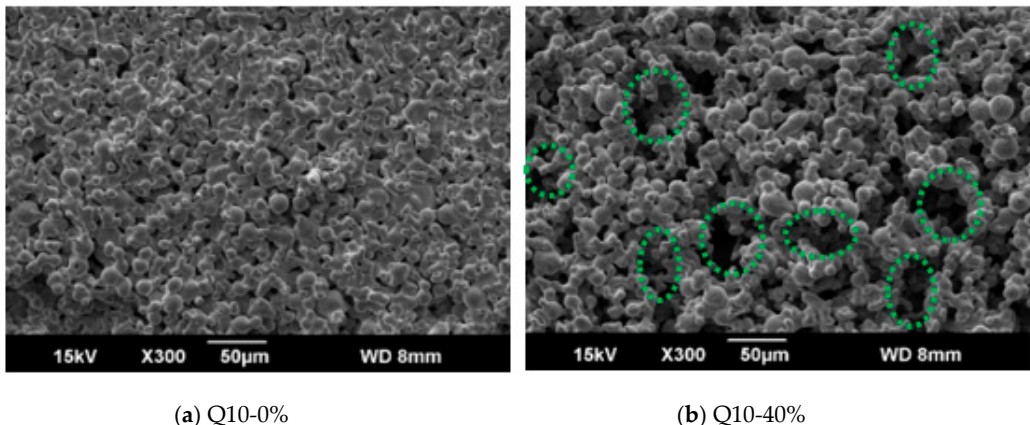

(**a**) Q10-0%          (**b**) Q10-40%

**Figure 5.** Electron micrographs of porous microchannels with pore-forming agent. Green dotted circle represent large void created by PFA.

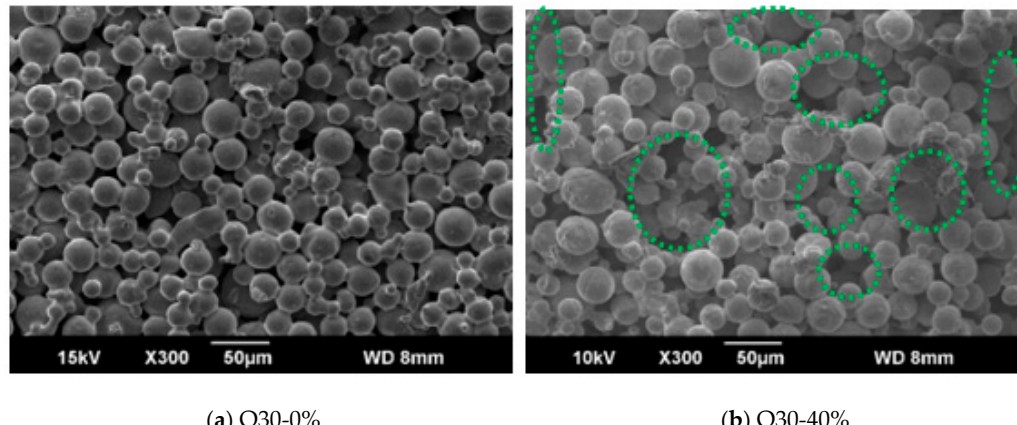

(**a**) Q30-0%                                                      (**b**) Q30-40%

**Figure 6.** Electron micrographs of porous microchannels with pore-forming agent. Green dotted circle represent large void created by PFA.

From Tables 2 and 3, the addition of pore-forming agent does increase the porosity of the sample. The porosity of Q10-0% is only 20.4%, while the porosity of the Q10-40% sample can reach 52.8%. The porosity of the Q30-40% sample is 47.5%.

**Table 3.** Parameters of microchannels sintered with 30 μm spherical copper powder.

| Particle Size | 30 μm | | | |
|---|---|---|---|---|
| specification | Q30-0% | Q30-10% | Q30-20% | Q30-40% |
| volume content | 0% | 10% | 20% | 40% |
| porosity | 26.4% | 33.2% | 40.4% | 47.5% |

## 3. Data Processing and Error Analysis

### 3.1. HTC Calculation

The effective heat flux of the test section is calculated by:

$$q_{eff} = \frac{Q_{eff}}{A_{base}} = \frac{Q_{eff}}{WL} \tag{1}$$

where $A_{base}$ is the top surface area of the copper block, W and L are the width and length of the microchannel sample, respectively, and $Q_{eff}$ is the effective heat input.

$$Q_{eff} = Q_{total} - Q_{loss} \tag{2}$$

where $Q_{total}$ is the total input heating power. $Q_{loss}$ is the heat loss between heat sink and environment.

The length of the single-phase section is given by [5]:

$$L_{sp} = \frac{\dot{m} \times c_p (T_{sat} - T_{in})}{q_{eff} \times W} \tag{3}$$

where $L_{sp}$ is the total length of the microchannel, m; $T_{sat}$ and $T_{in}$ are saturation enthalpy and inlet temperature of water, KkJ/kg, respectively. qis effective heat quantity, W/cm$^2$.

Then the length of the two-phase section:

$$L_{tp} = L_h - L_{sp} \tag{4}$$

The average measured temperature can be figured out from three thermocouples:

$$\overline{T_{tc}} = (T_{tc1} + T_{tc2} + T_{tc3})/3 \tag{5}$$

Then the surface temperature is calculated by the one-dimensional heat conduction model [7]:

$$T_w = \overline{T}_{tc} - q_{eff} \cdot \frac{y_{cop}}{\lambda_{cop}} \tag{6}$$

where, $q_{eff}$ is the effective heat flux, W/cm$^2$; $y_{cop}$ is the distance between the temperature measuring hole and upper surface of the copper plate; $\lambda_{cop}$ is the thermal conductivity of copper block, W/m·K.

The wall superheat is:

$$\Delta T_w = T_w - T_s \tag{7}$$

The average heat transfer coefficient ($h_{avg}$) is calculated by the weighted average of the single-phase HTC (hsp) and the two-phase boiling HTC ($h_{tp}$) [16]:

$$h_{tp} = q_{eff} / \left( T_w - \frac{T_{in} + T_{sat}}{2} \right) \tag{8}$$

$$h_{tp} = \frac{q_{eff}}{T_w - T_{sat}} \tag{9}$$

$$h_{avg} = \frac{h_{sp}L_{sp} + h_{tp}L_{tp}}{L_h} \tag{10}$$

### 3.2. Error Analysis and Heat Loss

Relative uncertainties of mass flux, inlet and outlet pressure and heating power (DC power supply) are ±1%, ±0.25% and ±0.5%, respectively. Inlet and outlet fluid temperatures were measured by two thermal resistance sensors (PT100) with an absolute uncertainty of ±0.3 K. Uncertainties of T-type thermocouples for measuring the copper block temperature are ±0.4 K. In terms of the error-transfer formula, the relative uncertainty of HTC is within the range of ±2.6~10.4%. The HTC value near the ONB point (Onset of Nucleate Boiling) usually presents a large relative uncertainty because its wall superheat may be even lower than 1.0 K for the porous microchannel. The relative uncertainty of the pressure drop is ±0.1%.

There exists thermal convection loss between test section chamber and environment. This heat loss in boiling condition is obtained by the extrapolation method from the single-phase flow test [17]. The heat loss is assumed proportional to the temperature difference in the copper block and the ambient. On the condition of $T_{in}$ = 60 °C, G = 142 kg/(m$^2$·s), the fitting curve of heat loss is as follows:

$$Q_{loss} = 1.07 \times (T_w - T_{sur}) - 37.9 \tag{11}$$

The thermal efficiency of the microchannel test section reaches 85~95% in terms of the heat loss curve.

Pressure effect on the saturation temperature is considered. Wall superheats of boiling curve were modified in terms of the followed relation between water saturation temperature and saturation pressure [17]:

$$T_{sat} = 0.256 \times p_{ave} + 73.859 \tag{12}$$

where pP$_{ave}$ represents the average pressure of microchannel inlet and outlet (kPa).

## 4. Results and Discussion

### 4.1. Boiling Curves and Heat Transfer Coefficients

Figure 7a shows the boiling curve of Q10-series samples. The effects of PFA content are exhibited. The inlet subcooling degree is 40 °C and the mass flux is 142 kg/m$^2$·s. During the experiment, three-point temperatures along the microchannel direction are measured. The average temperature is selected to calculate the wall superheat degree. The ONB

can be identified at the point with a radical change in slope of the boiling curve. At low heat fluxes lower than 20 W/cm², all curves almost coincide in single-phase state. When the heat flux is over 40 W/cm² boiling curves begin to differ greatly. Q10-20% presents the lowest wall superheats and the highest CHF of 160 W/cm² of all samples, showing excellent boiling performance. Other samples reach only about 120 W/cm². It indicates that too large or small porosity would degrade the boiling performance. Figure 7b shows boiling curves of Q30 series samples. Incipient wall superheats of all samples are less than 3.0 K. Q30-20% displays the lowest wall superheats and highest CHF of 160 W/cm² of all samples. Comparatively, the sample without PLA and larger percent PLA could not give full play to the best performance. Although sample Q30-10% can reach higher CHF, its wall superheats relatively more.

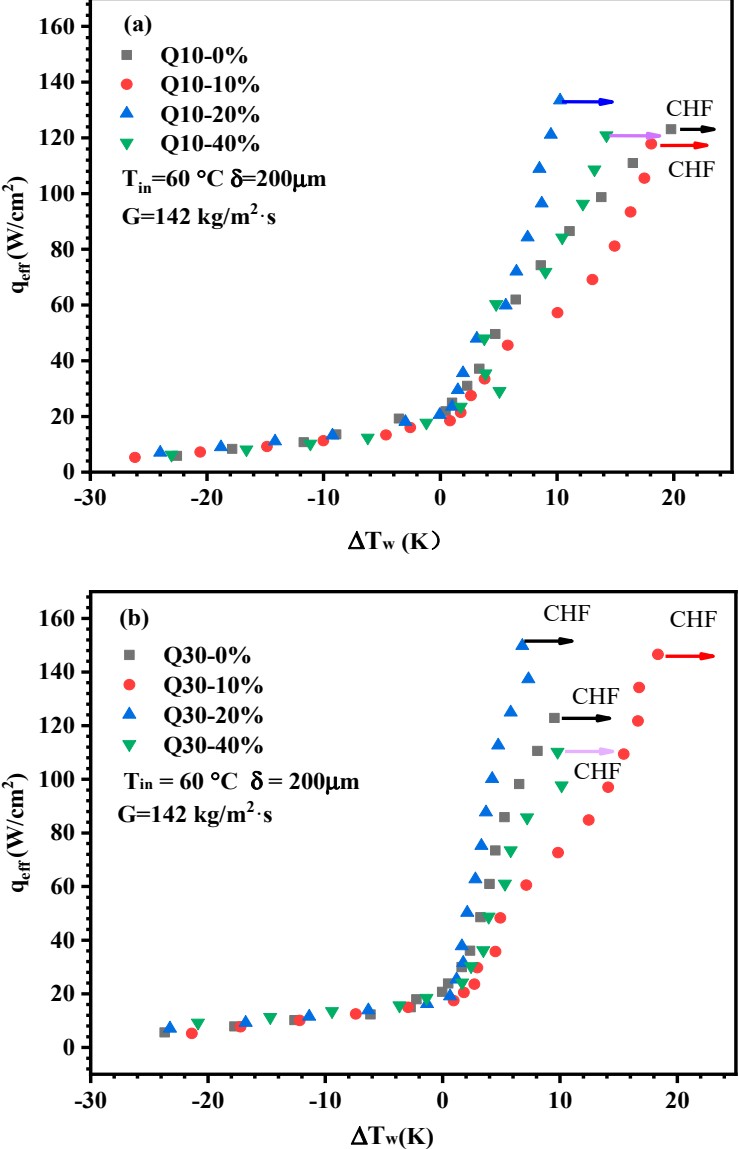

**Figure 7.** Boiling curves of porous-microchannel samples with different PFA contents. (**a**) Q10-series; (**b**) Q30-series.

The corresponding HTC curves of each sample are shown in Figure 8a,b. Heat transfer coefficients versus heat fluxes are illustrated in Figure 7a for the Q10 series samples. As the heat flux is less than 20 W/cm², HTCs show an almost constant single-phase state. After initiation, HTCs increase rapidly with heat fluxes. Q10-20%, with moderate PFA

content, achieves a higher HTC level. The maximum HTC of Q10-20% reaches up to 112.4 kW/(m²·K). However, for the sample without PLA, HTCs would decrease after moderate heat fluxes.

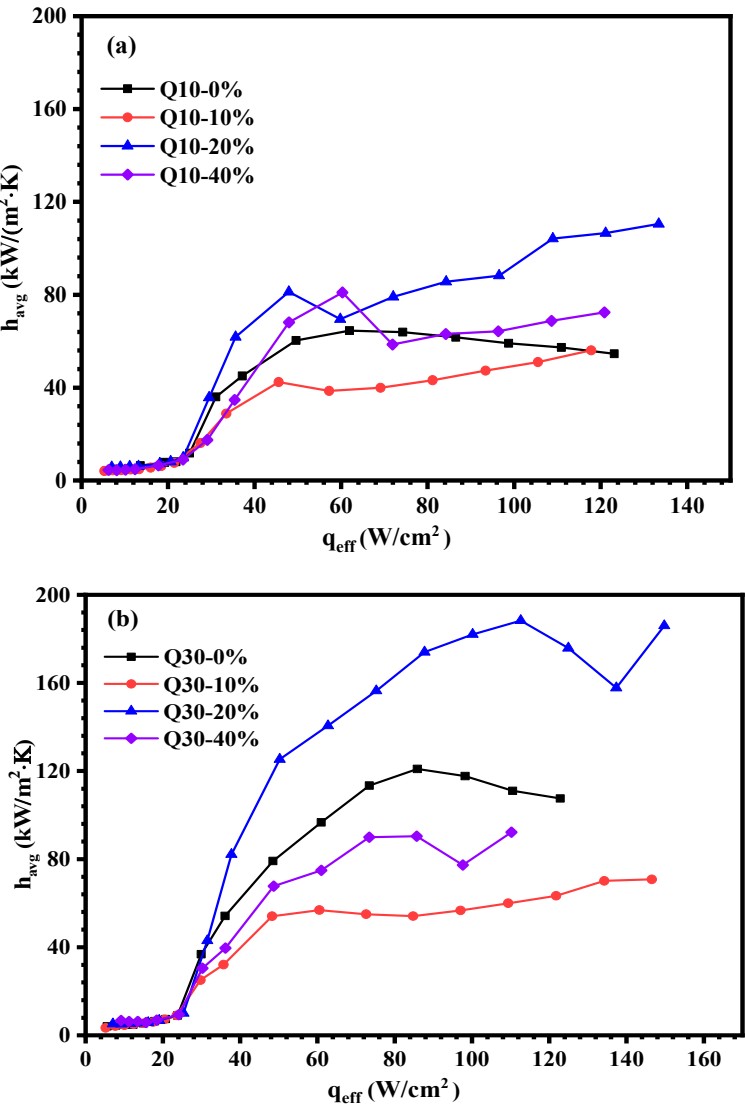

**Figure 8.** HTC curves of porous-microchannel samples with different PFA contents. (**a**) Q10-series; (**b**) Q30-series.

For Q30 series, the variation in HTC against heat flux is shown in Figure 7b. Compared with the Q10 series, Q30 series shows better heat transfer performance than Q10 series with the same PLA content. It demonstrates that the particle size effect also plays a vital role in the heat transfer performance, not only the porosity effect. Q30-20%, also moderate PLA content, achieves the highest in HTC of all samples and its maximum HTC reaches 182.7 kW/(m²·K). Comparatively, the maximum HTC of Q30-40% is only 98 kW/(m²·K).

The best boiling heat transfer performance is at a maximum for samples with medium PLA content. For Q10 series, the sample with 20% PLA presents superior performance and, for the Q30 series, the sample with the same content also does best. This suggests there exists an optimum porosity for porous microchannels. From the perspective of mechanism, low-porosity sample is not conducive to the vapor discharge and leads to earlier occurrence of film boiling while a large-porosity sample would reduce the evaporation area and effective thermal conductivity.

### 4.2. Average Pressure Drop

Figure 9a,b shows measured average pressure drops of Q10 and Q30 series samples. The overall trend of the average pressure drops increases with the increase in heat fluxes. The incipience point of the two-phase state is marked with a dotted line in the figure to distinguish the single-phase state. When the heat flux is less than 20 W/cm² all porous microchannels stay in the single-phase flow state. The pressure drop increases slightly with increasing heat flux. After the boiling incipience the average pressure drop is almost proportional to the heat flux. Q10-10% shows slightly higher pressure drop in each series. In general, the addition of PLA has a minor impact on the average pressure drop and would not cause a significant increase in the two-phase pressure drop.

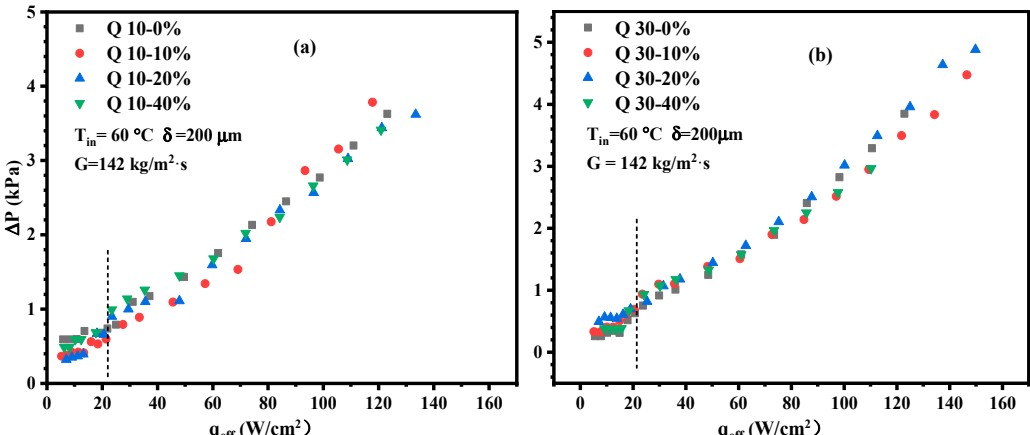

**Figure 9.** Average pressure drop curves of samples with different PFA contents. Dotted line represent boundary from single-phase to two-phase state. (**a**) Q10-series; (**b**) Q30-series.

### 4.3. Pressure Instability and Boiling Pattern of Sintered Microchannels Containing PLA

The PFA samples effectively alter sample porosities. Q10-series samples are taken as examples to further study the PLA effect on flow boiling. Figure 10 shows the pressure pulsation curve of Q10-0% sample near CHF at the heat flux of 121.2 W/cm². Inlet and outlet pressure presents an unstable large-amplitude, low-frequency oscillation type (LALF), superposed by small-amplitude, high-frequency type (SAHF). This is a typical pressure-oscillation mode found in porous microchannels sintered with small-sized copper particles (d $\leq$ 50 μm) [6]. Maximum amplitude of LALF oscillations yields 17.0 kPa and those of SAHF ones 2.0 kPa. One typical event is visually shown in Figure 11, selected from one range of pressure fluctuation curve in Figure 10. At t = $t_0$, both inlet and outlet pressure reach a peak value. A large vapor mass is found near the entrance. Inlet liquid could not flow into the channels and some regions have become drought. At t = $t_0$ + 24 ms, inlet pressure has been released and some liquid begins to rewet upper-side channels. Intense evaporation takes place in these channels. An explosive boiling emerges in the middle region and induces a small pressure spike in the pressure curve. At t = $t_0$ + 164 ms, liquid inlets change from upper side to low side. Due to strong evaporation the inflow liquid is rapidly transformed into water vapor, and the formed vapor mass at upper-side channels is rapidly expanded towards upstream. The inlet and outlet pressure drops to almost the lowest value. At t = $t_0$ + 235 ms, a violent explosive boiling occurs abruptly near the upstream region due to the high wall superheat (ΔT = 19.2 K). Its extensive influence hinders liquid inflow completely and disrupts the rhythmical flow pattern.

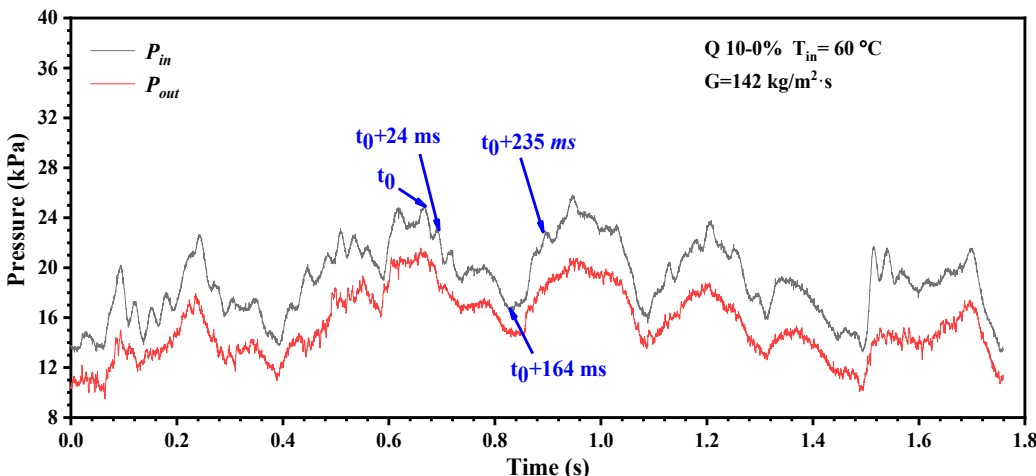

**Figure 10.** Pressure fluctuation curve of sample Q10-0% ($q_{eff}$ = 121.2 W/cm$^2$).

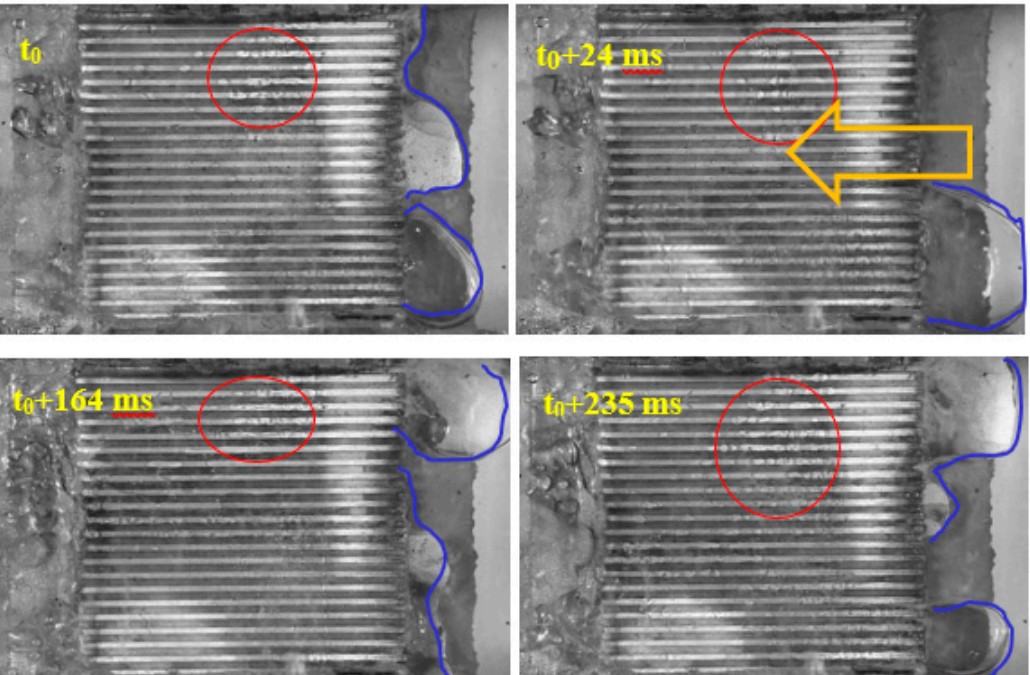

**Figure 11.** Visual picture of sample Q10-0% ($q_{eff}$ = 121.2 W/cm$^2$). Circle region: explosive boiling; Arrow: flow direction; blue line: vapor mass boundary.

Figure 12 shows pressure fluctuation at the inlet and outlet of the Q10-20% sample at G = 142 kg/(m$^2$·s) and q = 121.2 W/cm$^2$. Distinct from that of Q10-0%, the pressure curve of Q10-20% presents a quasi-periodicity. The addition of PLA with a volume ratio of 20% improves the flow instability. The corresponding visual observation is shown in Figure 13. At t = $t_0$, the inlet and outlet pressure reach the highest value, and inlet region is completely covered by a large vapor mass produced by evaporation, difficult for liquid to enter. At t = $t_0$ + 135 ms, the inlet liquid reenters those downside microchannels and rewets the relevant region. Both inlet and outlet pressure also fall to the trough. After t = $t_0$ + 198 ms, the liquid front has migrated to the middle region. Almost in the meantime, a new vapor pocket is growing up near the entrance, which leads to a new climb of inlet pressure. When t = $t_0$ + 305 ms the vapor pocket is completely formed. The inlet pressure reaches the crest and the inlet liquid was blocked outside.

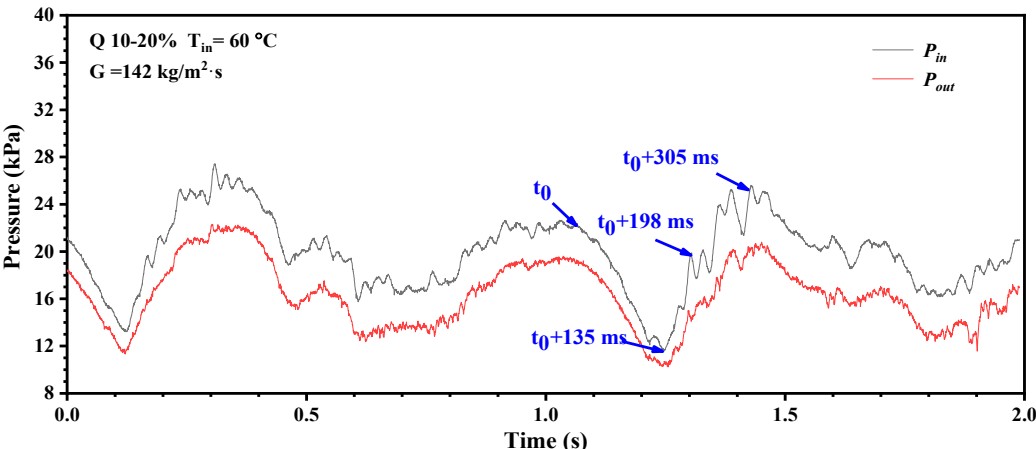

**Figure 12.** Pressure fluctuation curve of sample Q10-20% ($q_{eff}$ = 121.2 W/cm$^2$).

Thus, we can see that the addition of PFA improves pressure instability in sintered porous microchannels. Although there still exists some high-frequency fluctuation, it has a minor effect on flow periodicity. The oscillation amplitudes of both samples with PFA and no PFA range within 16.0 kPa.

Both inlet and outlet pressure curves of Q10-40% are shown in Figure 14. The pressure oscillation curves also show a quasi-periodicity. The oscillation amplitude is estimated to be 12.0 kPa, lower than that of Q10-20%. However, average duration of its pulsation cycle is only 0.35 s, substantially shorter than 0.53 s of Q10-20%. It means a more rapid evaporation-rewetting cycle for the sample of Q10-40%. It is speculated that large porosity may be a crucial contributing factor to this change, but this needs to be explored further.

### 4.4. Existence of Optimal PFA Content on CHF and Discussion

The PFA content also exerts a certain impact on the critical heat flux (CHF) of the sintering microchannel. Figure 14a,b reveals the correlation between the two. Figure 15a displays the CHF variation in Q10 series. Q10~20% achieves relatively higher CHF, more than 140 W/cm$^2$. Figure 15b shows the measured CHF values of Q30 series. Both Q30-10% and Q30-20% show a higher CHF of 160 W/cm$^2$. For both series, samples of 40% PFA content present a great decrease in CHF. This demonstrates that too large porosity is not conducive to the promotion of CHF.

The same layer thickness of 200 μm is for all samples in this study. A previous study [4] claimed that the ratio of layer thickness to particle size ($\delta/d$) had a mutual impact on the boiling on porous coatings, rather than by only particle size. The optimum $\delta/d$, acquired by pool boiling research [4], ranges from 2.0~4.0. The larger $\delta/d$, the worse the boiling heat transfer performance. The $\delta/d$ ratios of Q10 and Q30 yield 20.0 and 6.7, respectively. This is the reason why the Q30 series presents a better performance than the Q10 series. The addition of PFA increases sample porosity and creates the macropore, which greatly helps vapor discharge. In high heat fluxes the heating surface is completely covered by the vapor blanket. Smooth vapor discharge is a key factor to influence the CHF. However, with an increase in sample porosity CHF shows a decline trend after reaching a plateau. Visual observation combined with pressure fluctuation analysis discloses that the oscillation frequency is an important index. Larger porosity results in higher oscillation frequency that means a more rapid evaporation–rewetting cycle. It needs further exploration in the following research.

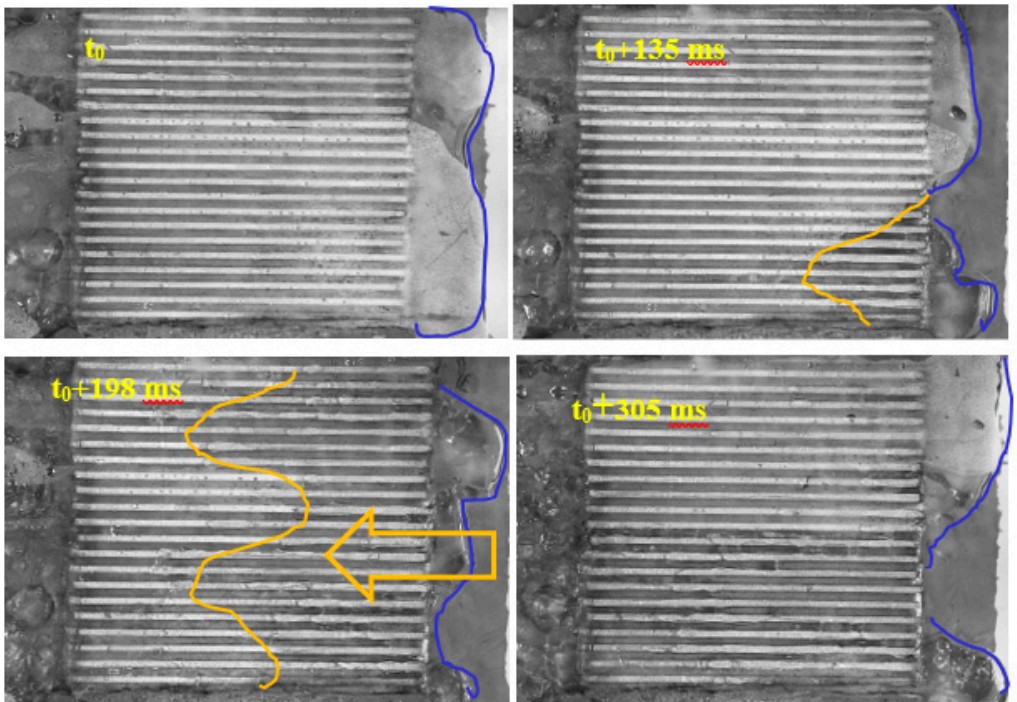

**Figure 13.** Visual images of sample Q10-20% ($q_{eff}$ = 121.2 W/cm²). Circle region: explosive boiling; Arrow: flow direction; blue line: vapor mass boundary.

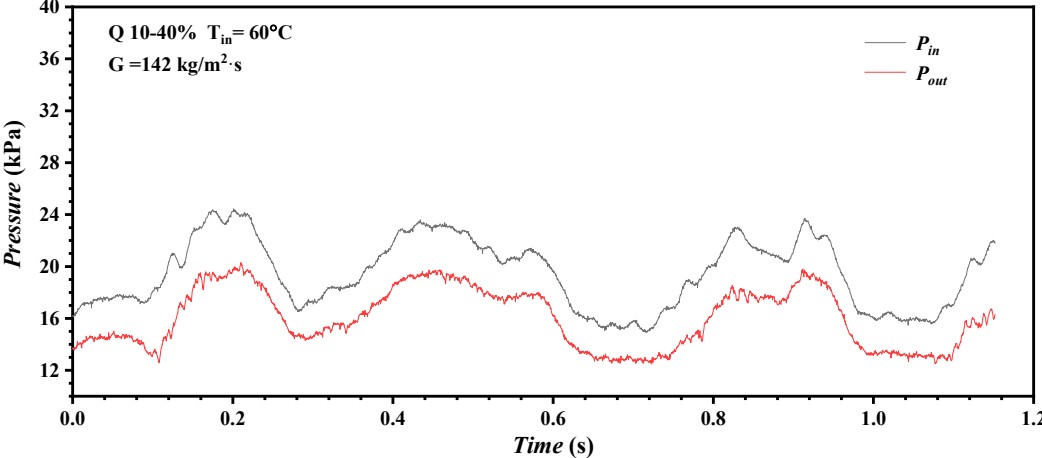

**Figure 14.** Pressure fluctuation curve of sample Q10-40% ($q_{eff}$ = 121.2 W/cm²).

Therefore, the combination parameter of δ/d presents a limitation in characterizing the boiling heat transfer performance. This comprehensive parameter could not consider the influence of other structural factors, such as porosity effect. Our results clearly show that porosity effect plays a pronounced role in the flow boiling process of porous microchannel.

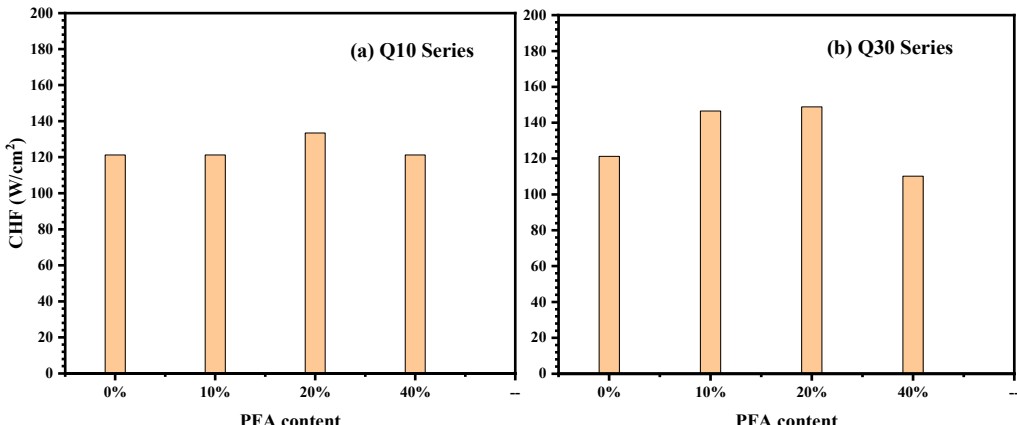

**Figure 15.** The relationship between PFA content and CHF for Q10 series and Q30 one. (**a**) Q10-series; (**b**) Q30-series.

## 5. Conclusions

In this work, the sintered microchannel samples with different PFA contents were systematically studied, and the conclusions were obtained as follows:

(1) For sintered parallel microchannels, the addition of PFA increases the sample porosity. Too large or too small content of PFA would degrade boiling heat transfer performance greatly. There exists an optimal range of PFA content for sintered microchannels. At the moderate PFA content, the boiling HTC of sintering microchannel reaches the maximum;

(2) PFA content has little effect on the average pressure drop and would not cause the rapid increase in flow resistance;

(3) According to the visual observation, the sample with moderate porosity is helpful to establish the rhythmic boiling and reduce the occurrence of explosive boiling. Larger porosity results in more rapid evaporation-rewetting cycle;

(4) Particle size also has a great influence on boiling curves and HTC ones. Q30 series could achieve better performance than Q10 series. This is attributed to larger ratio of layer thickness to particle size ($\delta/d$) for Q10-series samples.

The PFA content has a certain influence on the flow boiling process in the sintering microchannel, especially critical heat flux (CHF). Figure 14a,b reveals the correlation between the two. Figure 14a displays the CHF variation in Q10 series. Q10-20% achieves relatively higher CHF, more than 140 W/cm$^2$. Figure 14b shows the measured CHF values of Q30 series. Both Q30-10% and Q30-20% show a high CHF of 160 W/cm$^2$. For both series, samples of 40% PFA content present a great decrease in CHF. This demonstrates that too large porosity is not conducive to the promotion of CHF.

**Author Contributions:** Q.L., L.F. and J.M. contribute to experiment conduction. D.Z. contributes to overall design of this research; D.C. contributes to the analysis of the results, and conclusion. All authors have read and agreed to the published version of the manuscript.

**Funding:** This research was funded by International Science and Technology Cooperation Project of the Ministry of Science and Technology (G2022014065L).

**Data Availability Statement:** The data presented in this review are available from the corresponding author upon reasonable request.

**Acknowledgments:** We gratefully acknowledge the financial support of the Sino-Russian Joint Laboratory Project.

**Conflicts of Interest:** The authors declare no conflict of interest.

## Nomenclature

| | | | | |
|---|---|---|---|---|
| $A_{base}$ | top surface area of copper block, cm$^2$ | | $PFA$ | pore-forming agent |
| $CHF$ | critical heat flux, W/cm$^2$ | | $q_{eff}$ | effective heat flux, W/cm$^2$ |
| $d$ | particle diameter, mm | | $Q_{eff}$ | effective input heat power, W |
| $G$ | mass flux, kg/m$^2$·s | | $Q_{loss}$ | heat loss, W |
| $HTC$ | heat transfer coefficient | | $Q_{total}$ | input heat power, W |
| $h_{avg}$ | average heat transfer coefficient, kW/(m$^2$·K) | | $T_w$ | wall temperature, K |
| $h_{sp}$ | single-phase heat transfer coefficient, kW/(m$^2$·K) | | $T_{in}$ | inlet temperature, K |
| $h_{tp}$ | two-phase heat transfer coefficient, kW/(m$^2$·K) | | $T_s$ | saturation temperature, K |
| $L$ | length of microchannel (or sample), cm | | $y_{cop}$ | distance, mm |
| $L_{sp}$ | two-phase length of microchannel, cm | | | |
| $L_{tp}$ | single-phase length of microchannel, cm | | Greek symbols | |
| $\dot{m}$ | mass flow rate, kg/s | | $\Delta P$ | pressure drop, kPa |
| $N$ | microchannel number | | $\lambda_{cop}$ | thermal conductivity, W/(m·K) |
| $ONB$ | onset of nucleate boiling | | $\delta$ | bottom layer thickness, mm |
| $P_{ave}$ | average pressure, kPa | | | |
| $P_{in}$ | inlet pressure, kPa | | | |
| $P_{out}$ | outlet pressure, kPa | | | |

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
