# Peer review of "Boiling Heat Transfer Characteristics of Porous Microchannel with Pore-Forming Agent"

_processes, doi:10.3390/pr11020617_

Round 1

Reviewer 1 Report

The manuscript "Boiling heat transfer characteristics of porous microchannel with pore-forming agent " performed a basic performance analysis of porous microchannels using a pore-forming agent (PFA) for flow boiling experiments. The reported results showed that the boiling heat transfer coefficient attained its maximum at moderate porosity. The article is interesting; however, I have the following observations that should be addressed in letter and spirit before consideration of the manuscript for publication.

1.      The authors have used a few abbreviations (PFA, Q10, Q30) in the abstract section; however, these abbreviations are not defined at their first appearance. Authors should define all the abbreviations used in the text's first appearance.

2.      In the abstract section, the authors mentioned the effect of PFA on the porosity, boiling heat transfer coefficient and average pressure drop. However, the percentage change effect with and without using PFA on these parameters in terms of numerical values should also be reflected in the abstract section instead of just elaborating on an increase or decrease.

3.      The keywords should be arranged alphabetically, and authors are desired to pick the most relevant and understandable keywords specific to the present research work.

4.      The "introduction section" is too short and lacks the current work's novelty in specific application areas. The authors have cited multiple research studies; however, the literature's critical contribution and outcomes need more elaboration. There is also a dire need to explore the critical parameters that have not been considered in the cited literature and develop a statement of the novelty of the present work in the context of reported studies.

5.      Authors should cite more recent research studies in the presented research scope, elaborate on the scope of the present work and strengthen the novelty. This will also help expand the overall content of the "introduction section". The last paragraph elaborating on the "objectives of the current study is also weak and not well described.

6.      In section 2.1, the authors have described different components used in the experimental system. However, the authors did not mention the specifications, limitations, or uncertainties of the instruments used. The authors should mention all of these, along with the type and models of the critical instruments used. Table 1 should also be placed in section 2.1.

7.      The quality of Figure 1 is low, and the aspect ratio of the text is improper. Therefore, authors are advised to improve the flow diagram by considering the aesthetics using different colour combinations. Also, the names mentioned against each number in the "Figure Captions" should be placed separately in a tabular form.

8.      Although Figure 1 reflects the flow diagram of the experimental setup, authors are also advised to add the actual picture of the experimental setup used in this study. Also, Figure 2 should be appropriately labelled.

9.      The authors should mention the model and specifications of the scanning electron microscope (SEM) used for comparing the copper powder samples. In addition, critical discussion on the results obtained from the SEM images is also not well elaborated and should be added.

10.  The authors should adequately define all the abbreviations used in equations 1 to 11. It was also observed that for most of the correlations reported (equations 1-11), the authors did not mention the proper reference from where they picked these mathematic models. Authors should cite the literature for each of the correlations used.

11.  In Figures 6-9, 11, 13 and 14, the authors have discussed the results obtained in each module; however, the science behind each process and critical discussion of the results is not adequately elaborated. In addition, the correlation and comparison of the presented results with the existing literature are strongly needed.

12.  The quality of Figures 10 and 12 is not good; also, the text used in the Figures is not in line with the text style and size present in the other manuscript figures. The critical analysis and discussion of the results obtained from the Figures is also missing

13.  The conclusion section is very weak, and the authors have replicated only most of the information from the results and discussion section. However, the conclusive, critical remarks gathered from the present study are not reflected and should be recorded.

Overall, the concept presented in the manuscript is exciting but lacks novelty, which should be addressed by critically analysing the existing literature in the present field. Furthermore, the English language of the manuscript is very poor, and there is a dire need to check the language and grammar of the manuscript. Therefore, the authors are desired to proofread the manuscript by an expert from a native English language-speaking country.

Author Response

1.The authors have used a few abbreviations (PFA, Q10, Q30) in the abstract section; however, these abbreviations are not defined at their first appearance. Authors should define all the abbreviations used in the text's first appearance.

Reply: This problem has been corrected. The PFA is defined in the abstract. The definitions of Q10 and Q30 are a little trivial and has been illustrated in Section 2.2: Q10 means the samples sintered with the copper particle size of 10 μm and Q30 means the ones with the copper particle size of 30 μm

2.In the abstract section, the authors mentioned the effect of PFA on the porosity, boiling heat transfer coefficient and average pressure drop. However, the percentage change effect with and without using PFA on these parameters in terms of numerical values should also be reflected in the abstract section instead of just elaborating on an increase or decrease.

Reply: The percentage change effect with and without using PFA has been listed in the abstract.

3.The keywords should be arranged alphabetically, and authors are desired to pick the most relevant and understandable keywords specific to the present research work.

Reply: OK. We consider keywords as follows: Flow boiling; Microchannel; Pressure fluctuation; Pore-forming agent; Sintering.

  1. The "introduction section" is too short and lacks the current work's novelty in specific application areas. The authors have cited multiple research studies; however, the literature's critical contribution and outcomes need more elaboration. There is also a dire need to explore the critical parameters that have not been considered in the cited literature and develop a statement of the novelty of the present work in the context of reported studies.

Reply: The pore-forming agent method was mostly applied in capillary-wick fabrication of heat pipe, but less investigated on flow boiling field. The related literature’s contribution has been elaborated in the revised paper. From the existing literatures, porosity effect on boiling performance was not well understood for porous structure. Our work selected the porous microchannel as a subject to explore the porosity effect in flow boiling process. For the porous microchannel, the sample performance is influenced by various structural parameters: particle size, porosity, and pore morphology, etc. These parameters are coupled together, so it is a complicated problem. In Section 1, paragraph 3, the novelty of the present work has been elucidated and expanded.

5.Authors should cite more recent research studies in the presented research scope, elaborate on the scope of the present work and strengthen the novelty. This will also help expand the overall content of the "introduction section". The last paragraph elaborating on the "objectives of the current study is also weak and not well described.

 Reply: More recent work has been expounded in the revised paper. The last paragraph has also highlighted the objectives of the current study.

6.In section 2.1, the authors have described different components used in the experimental system. However, the authors did not mention the specifications, limitations, or uncertainties of the instruments used. The authors should mention all of these, along with the type and models of the critical instruments used. Table 1 should also be placed in section 2.1.

Reply: The specifications, limitations, or uncertainties of some instruments has been illustrated in revised manuscript. Table 1 has moved to Section 2.1, and related description has also  been supplemented.

7.The quality of Figure 1 is low, and the aspect ratio of the text is improper. Therefore, authors are advised to improve the flow diagram by considering the aesthetics using different colour combinations. Also, the names mentioned against each number in the "Figure Captions" should be placed separately in a tabular form.

Reply: The quality of Figure 1 has been adjusted and improved. Different colors are indicated in this figure: Red color represents fluid unit and blue color represents measurement unit.

The names mentioned is plotted directly in the "Figure Captions".

8.Although Figure 1 reflects the flow diagram of the experimental setup, authors are also advised to add the actual picture of the experimental setup used in this study. Also, Figure 2 should be appropriately labelled.

Reply: Due to limited time, this paper will not present the actual picture in the flow diagram, but it’s a good suggestion. Figure 2 has also been labelled afresh.

9.The authors should mention the model and specifications of the scanning electron microscope (SEM) used for comparing the copper powder samples. In addition, critical discussion on the results obtained from the SEM images is also not well elaborated and should be added.

Reply: SEM specification has been mentioned. The discussion about microstructures of SEM images is also elaborated in detail.

10.The authors should adequately define all the abbreviations used in equations 1 to 11. It was also observed that for most of the correlations reported (equations 1-11), the authors did not mention the proper reference from where they picked these mathematic models. Authors should cite the literature for each of the correlations used.

Reply: All the abbreviations in equations 1 to 12 has been defined in Nomenclature. All equations are reorganized again. Equation 8 is an important expression referred from [14].

11.In Figures 6-9, 11, 13 and 14, the authors have discussed the results obtained in each module; however, the science behind each process and critical discussion of the results is not adequately elaborated. In addition, the correlation and comparison of the presented results with the existing literature are strongly needed.

Reply: These problems do exist. However, porosity effect on the flow boiling process was rarely

investigated in the porous layer, especially not to mention in porous microchannel.   

From existing research, the PFA sintering method was applied almost entirely in Looped heat pipe. This work made first attempt to apply this method in the flow boiling process of porous microchannel. It focused mainly on the performance influence of PFA content to acquire the porosity effect on boiling performance. Visual observation, combined with pressure instability analysis, also was put forward to explore the boiling and evaporation process. It's true that mechanism illustration is not completely satisfied, but it sheds light on further study.

Comparison of the presented results is not shown because of the scarcity of relevant research (Flow boiling in the porous microchannel sintered by spherical copper particles of

d =10µm or 30 µm ).

12.The quality of Figures 10 and 12 is not good; also, the text used in the Figures is not in line with the text style and size present in the other manuscript figures. The critical analysis and discussion of the results obtained from the Figures is also missing.

Reply:  The quality of Figures 10 and 12 has been improved. The analysis and discussion of

two Figures is shown in the Paragraph 1 and 2 of the Section 4.3.

13.The conclusion section is very weak, and the authors have replicated only most of the information from the results and discussion section. However, the conclusive, critical remarks gathered from the present study are not reflected and should be recorded.

Reply:  The conclusion section has been improved.

Overall, the concept presented in the manuscript is exciting but lacks novelty, which should be addressed by critically analysing the existing literature in the present field. Furthermore, the English language of the manuscript is very poor, and there is a dire need to check the language and grammar of the manuscript. Therefore, the authors are desired to proofread the manuscript by an expert from a native English language-speaking country.

Reply: English writing has been improved.

Reviewer 2 Report

The authors presented an experimental investigation on the Boiling heat transfer characteristics of porous microchannel with pore-forming agent

The paper can be accepted for publication after minor revision:

Some quantitative findings are to be added to the introduction.

The introduction is very short and is to be extended.

The novelty of the work is to be clearly stated.

An actual photo of the experimental setup is to be added.

Figure 3, has low resolution

The data acquisition system is to be described in detail.

An experimental uncertainty study is to be performed, it is not sufficient to present only the uncertainty of the measurement devices. It mandatory to present the uncertainty of the evaluated results (hav, pressure drop….).

In figure 10, there is a Chinese word on the yellow arrow. The same for fig 14.

Why the heat flux is limited to 16 w/cm2?

The discussion is to be improved by adding physical interpretations.

It will be interesting to evaluate the thermal thermal–hydraulic performance factor (TPF).

Author Response

The authors presented an experimental investigation on the Boiling heat transfer characteristics of porous microchannel with pore-forming agent

The paper can be accepted for publication after minor revision:

Some quantitative findings are to be added to the introduction.

Reply: Some quantitative findings has been added to the introduction.

The introduction is very short and is to be extended.

Reply: More recent work has been expounded in the revised paper.

The novelty of the work is to be clearly stated.

Reply:  The last paragraph has also highlighted the objectives of the current study. The pore-forming agent method was mostly applied in capillary-wick fabrication of heat pipe, but less investigated on flow boiling field. The related literature’s contribution has been elaborated in the revised paper. From the existing literatures, porosity effect on boiling performance was not well understood for porous structure. Our work selected the porous microchannel as a subject to explore the porosity effect in flow boiling process. For the porous microchannel, the sample performance is influenced by various structural parameters: particle size, porosity, and pore morphology, etc. These parameters are coupled together, so it is a complicated problem. In Section 1, paragraph 3, the novelty of the present work has been elucidated and expanded.

.An actual photo of the experimental setup is to be added.

Reply: Due to limited time, this paper will not present the actual picture in the flow diagram, but it’s a good suggestion. In the following papers, we would consider the actual picture in the flow diagram.

Figure 3, has low resolution

Reply: we have improved the resolution.

The data acquisition system is to be described in detail.

Reply: DAC system has been elaboration afresh in the revised manuscript.

An experimental uncertainty study is to be performed, it is not sufficient to present only the uncertainty of the measurement devices. It mandatory to present the uncertainty of the evaluated results (hav, pressure drop….).

Reply: Uncertainty analysis has been carefully considered during the study.

It is never too exaggerated to emphasize that this flow boiling experiment is closely related to uncertainty control, including heat loss, wall temperature and flow rate uncertainty.  The pressure drop uncertainty is determined by pressure transducers, PX309-015G5V, which is about ±0.1%.

In figure 10, there is a Chinese word on the yellow arrow. The same for fig 14.

Reply: Sorry, it has been deleted.

Why the heat flux is limited to 16 w/cm2?

Reply: The reason is that HTCs in single-phase flow state(qeff≤16 W/cm2) almost keeps constant.

The discussion is to be improved by adding physical interpretations.

Reply: The discussion has been improved.

It will be interesting to evaluate the thermal thermal–hydraulic performance factor (TPF).

Reply: Good suggestion, Due to time limitation, TPF would be evaluated in the following work.

Reviewer 3 Report

Author have reported boiling heat transfer study through experiental work. Experimental results showed that the addition of pore-forming agent (PFA) could increase the sample porosity. At the moderate porosity, the boiling heat transfer coefficient (HTC) reached the maximum for both Q10 and Q30 series. Too large or too small porosity would degrade boiling heat transfer performance. It demonstrated there existed an optimal range of PFA content for sintered microchannels. PFA content has a minor effect on the average pressure drop and would not cause the rapid increase of flow resistance. Visual observation disclosed that the sample porosity would affect the pressure instability significantly. The sample with moderate porosity showed periodic pressure fluctuation and could establish rhythmical boiling. Particle size also exerted a certain influence on the boiling heat transfer performance. Q30 series could achieve higher HTC and CHF than Q10 series. Keywords: Pore-forming agent; sintering; porous microchannel; flow boiling; pressure fluctuation 1. Introduction The problem of heat dissipation has become the main obstacle of product upgrading. The heat flux in local hot spot exceeds even more than 2

General comments :

Author may rewrite the conclusion and introduction section for better understanding.

Author may add some recent research/review articles on boiling heat transfer to make study more relevent and add some novelty of work and and compare the current study with the existing one. 

Author may add some relevent research 

some examples are below:
10.3390/en15155759

Author may give his/her comments of surface rougness value and effect of contact angle on the boiling study.

Author Response

Author may rewrite the conclusion and introduction section for better understanding.

Reply: The conclusion and introduction section has been improved.

Author may add some recent research/review articles on boiling heat transfer to make study more relevent and add some novelty of work and and compare the current study with the existing one. 

Author may add some relevent research 
Reply: More recent work has been expounded in the revised paper. The last paragraph has also highlighted the objectives of the current study.

some examples are below:
10.3390/en15155759

https://doi.org/10.1080/01457632.2022.2086095

Author may give his/her comments of surface rougness value and effect of contact angle on the boiling study.

Reply: This paper focuses on boiling Heat Transfer on cylindrical Surface. It’s a little far away from the subject of porous microchannel. If you are interested, maybe we discuss through email.

My email is: [email protected].

Round 2

Reviewer 1 Report

Authors are advised to address the following observation as presented before in the review report

" It was also observed that for most of the correlations reported (equations 1-11), the authors did not mention the proper reference from where they picked these mathematic models. Authors should cite the literature for each of the correlations used."

Author Response

Reply: Sorry. I originally consider most of equations are widely used in boiling heat transfer field. Now some of equations has been cited in the new manuscript.

Reviewer 2 Report

it is mandatory to add an actual photo of the experimental setup; it is necessary for the credibility of the work

Author Response

Reply: Sorry, a new experimental setup figure is added in the revised paper.

Reviewer 3 Report

Thank you for your afforts in revising the manuscript. I doubt still author has not revised entire manuscript properly.

some points are mentioned below:

Please pay attention :

Nomenclature table is not properly organized.
figure 1 resolution is very poor.
you have used short form of CHF without full form (please keep non subject expert also into your kinsideration). author may read atleast to avoid such mistakes in manuscript before submission in such a reputed journal. 

I was very surprised that out of cited references there is no single reference to any papers published in processes journal. If the author did not find any references in this journal, that probably means that the paper is out of scope.  It is recommended that the author should do a much better job of reviewing the papers published in this journal on the general topic of this paper.  This would provide the readers a sense of continuity and help them pacing your paper in the context of what the journal has been publishing, very much strengthening your article's impact. This issue may be properly addressed.

Author Response

Reply:  Nomenclature table has been organized again in the revised manuscript.

       Figure 1 resolution has been improved.

       CHF has been expressed in abstract.

       In cited references, we replace reference [2] with a related paper from Processes.

I also search the published papers in processes and found less related ones. Sorry for that

Round 3

Reviewer 2 Report

after revision, the paper can be accepted for publication

Reviewer 3 Report

Paper can be accepted after editor's approval